# The Influence of Cryogenic Treatment on the Microstructure and Mechanical Characteristics of Aluminum Silicon Carbide Matrix Composites

**DOI:** 10.3390/ma16010396

**Published:** 2023-01-01

**Authors:** Mingli Zhang, Ran Pan, Baosheng Liu, Kaixuan Gu, Zeju Weng, Chen Cui, Junjie Wang

**Affiliations:** 1CAS Key Laboratory of Cryogenics, Technical Institute of Physics and Chemistry, Beijing 100190, China; 2University of Chinese Academy of Sciences, Beijing 100049, China; 3AVIC Manufacturing Technology Institute, Beijing 100024, China

**Keywords:** 15%SiCp/2009 aluminum matrix composite, cryogenic treatment, thermoelectric power, aging behavior, tensile property

## Abstract

Aluminum matrix composites have been widely used in aerospace and automotive fields due to their excellent physical properties. Cryogenic treatment was successfully adopted to improve the performance of aluminum alloy components, while its effect and mechanism on the aluminum matrix composite remained unclear. In this work, the effects of cryogenic treatment on the microstructure evolution and mechanical properties of 15%SiCp/2009 aluminum matrix composites were systematically investigated by means of Thermoelectric Power (TEP), Scanning Electron Microscopy (SEM) and Transmission Electron Microscopy (TEM). The results showed that TEP measurement can be an effective method for evaluating the precipitation characteristics of 15%SiCp/2009 aluminum matrix composites during aging. The addition of cryogenic treatment after solution and before aging treatment promoted the precipitation from the beginning stage of aging. Furthermore, the aging time for the maximum precipitation of the θ″ phase was about 4 h advanced, as the conduction of cryogenic treatment accelerates the aging kinetics. This was attributed to the great difference in the linear expansion coefficient between the aluminum alloy matrix and SiC-reinforced particles, which could induce high internal stress in their boundaries for precipitation. Moreover, the lattice contraction of the aluminum alloy matrix during cryogenic treatment led to the increase in dislocation density and micro defects near the boundaries, thus providing more nucleation sites for precipitation during the aging treatment. After undergoing artificial aging treatment for 20 h, the increase in dispersive, distributed precipitates after cryogenic treatment improved the hardness and yield strength by 4% and 16 MPa, respectively.

## 1. Introduction

In the past half century, the use of SiC particles as a reinforcing phase for the improvement of the machining and mechanical properties of aluminum alloys has been widely investigated [1,2,3]. When compared with the corresponding aluminum alloy, a particle-reinforced aluminum alloy matrix composite has a higher tensile strength [4,5], hardness [6] and fatigue strength [6]; qualities which have been widely used in the aerospace and automotive fields [7]. Similar to aluminum alloys, aluminum matrix composites can be subjected to solution and aging treatments for the enhancement of the mechanical properties of the material via solution and precipitation strengthening effects [8]. The aging behavior of aluminum matrix composites is quite different from that of the corresponding aluminum alloys due to the presence of the reinforced-particle phase. It was reported that the addition of SiC particles accelerated the formation of the S′ phase (Al_2_CuMg) in a SiCp-reinforced 2009 aluminum alloy [9]. However, research from Pal et al. [10] showed that it took a longer time for SiCp/Al-Cu-Mg composites to reach the peak of aging than it did for the corresponding alloy. Although SiC could affect the precipitation behavior and improve the mechanical properties of the composite’s material, it would not change the aging sequence in an aluminum matrix when compared with the corresponding aluminum alloy [11]. Therefore, solution and aging treatment can also be the effective methods to modify the comprehensive properties of an aluminum alloy and its composite [12].

Cryogenic treatment, which is the process of subjecting materials to a low-temperature environment, has been combined with solution and aging treatment to improve the comprehensive performances of aluminum alloys. Steier et al. [13] found that deep cryogenic treatment promoted the formation of additional GP-zones in 6101 aluminum alloy as a result of improving the wear resistance. A deep cryogenic treatment with a long soaking time (45 h) had positive effects on the mechanical properties and machinability of a 5083 alloy [14]. Gao et al. [15] also found that deep cryogenic treatment improved the strength and plasticity of a 7A99 alloy through refining the precipitates and increasing the uniformity of the precipitate distribution. Park [16] revealed that cryogenic treatment followed by rapid reheating (uphill quenching) released the residual stress and obviously improved the tensile properties. It has been reported by Araghchi [17] and Wang et al. [9] that cryogenic treatment can promote the formation of small-size precipitates to improve the mechanical properties. It was first reported by Jovičević-Klug [18] et al. that deep cryogenic treatment facilitated the precipitation of the β″ phase while suppressing the formation of the β′ phase with a large size during artificial aging treatment. Zhou et al. [19] attributed the improvement of aluminum alloys to the reduction of residual stress and a balanced state of residual stress in the material. Araghchi et al. [17] claimed a promotion of nucleation sites for the S′ phase which resulted from the increase in dislocation density following cryogenic treatment in a 2024 aluminum alloy.

Owing to the positive effects of cryogenic treatment on aluminum alloys, investigation on cryogenically treated aluminum matrix composites has aroused concerns for both academia and industry in recent years. Li et al. [20] investigated the effects of cryogenic treatment on Al_2_O_3_/Al-Zn-Mg-Cu matrix composites and found that precipitation of the S phase was facilitated at −108 °C when samples were recovered from cryogenic temperature to room temperature. Hong [21] et al. claimed that cryogenic treatment conducted in combination with aging for aluminum matrix composites promoted the formation of the S′ phase around SiC particles and further improved their mechanical properties. The effects of cryogenic treatment on the aging behavior of aluminum matrix composites were further investigated by Wang [9] et al., who showed that deep cryogenic treatment induced the generation of thinner θ′ precipitates in the grains with the increase in artificial aging time. It can be seen that the effects of cryogenic treatment have a close relationship with the subsequent aging parameters in aluminum matrix composites. However, the in-depth mechanisms surrounding the effects of cryogenic treatment on the microstructure evolution during subsequent aging in aluminum matrix composites remain unrevealed.

It can be concluded that the changes in precipitation characterizations are the main mechanism for the cryogenic treatment of aluminum alloys and composites and have great relationships with the type of alloys, the heat treatment parameters, and so on. However, the precipitation in aluminum alloys and composites induced by cryogenic treatment is not as obvious as the precipitation induced by aging treatment alone. The method of conventional microstructure characterization can be easily influenced by the selected areas and have a deficiency in the overall analysis of the microstructure evolution. Thermoelectric Power (TEP) is an intrinsic property of material which is related to electron scattering caused by the existence of crystalline defects such as the displacement of ions by thermal oscillations, the presence of accidental impurities or solute atoms, and imperfections including dislocations, grain boundaries, stacking faults, vacancies, and so on [22]. Therefore, a TEP measurement can be used for the precise and global evaluation of a microstructure evolution due to its microstructure sensitivity. Bakare et al. [23] revealed the formation behavior of GP-zones and the changing trend of dislocations with the increase in aging time in aluminum alloys by TEP measurement. Abdala et al. [24] found that TEP measurements were very sensitive to precipitation phenomena in a 6013 alloy, which allowed for the proposal of a precipitation sequence for this alloy. Massardier et al. [25] demonstrated that the TEP kinetics allowed an assessment of the residual concentration of solute in the final equilibrium state. However, investigation into the microstructure evolution of aluminum matrix composites induced by cryogenic treatment via TEP measurements and analysis has not been brought to the forefront.

Therefore, the present work aims to investigate the effects of cryogenic treatment on the microstructure changes and mechanical properties of 15%SiCp/2009Al aluminum matrix composites in different aging stages. Use of TEP measurements in conjunction with conventional microstructure characterization approaches are adopted for a global and systematical evaluation of the microstructure evolution. The influence of the combination sequence among solution, cryogenic, and aging treatments on the mechanical properties of aluminum matrix composites is also studied.

## 2. Materials and Methods

The aluminum matrix composite used in this study was the 15 vol% silicon-carbide-particle-reinforced 2009Al (SiCp/2009Al) composite. The material was provided in an as-forged state by the Institute of Metal Research of the Chinese Academy of Sciences (IMR). Its chemical composition is shown in Table 1.

A solution treatment (S) was carried out using a SX-5-12 box resistance furnace. The samples were heated to 510 °C and held for 2 h with subsequent water-quenching at room temperature. Cryogenic treatment is usually divided into shallow cryogenic treatment (−40~−80 °C) and deep cryogenic treatment (−80~−196 °C) according to the minimum temperature [14,26]. The previous research has demonstrated that the improving effects of cryogenic treatment can be promoted with a decrease in treating temperature [27]. In this work, cryogenic treatment (C) was conducted by cooling down quenched samples to −196 °C at the rate of 1 °C/min. The samples were then held for 12 h. The program-controlled SLX-80 cryogenic system was used for carrying out cryogenic treatment, as is shown in Figure 1. After 12 h of storage, samples were warmed up to room temperature in the atmosphere for 15 min. The artificial aging treatment (A) was carried out using a DZF-6210 vacuum-drying oven. During the aging process, the composite materials were heated to 170 °C and held for r2, 6, 10, and 20 h, respectively, before being cooled to room temperature. The time of cooling to room temperature was approximately 10 min. In this study, cryogenic treatment was conducted in combination with solution and aging treatments in various sequences. All combinations are presented with necessary illustrations in Table 2. The alphabetic order of each combination refers to the specific conducting sequence of the three basic treatments. For example, the process of SCA6 indicated that the samples were subjected to solution treatment, cryogenic treatment, and aging treatment for 6 h successively.

The thermoelectric power (TEP) method was adopted for evaluating the microstructure evolution during aging. The principle of TEP measurement was to establish a temperature gradient at both ends of the samples and to measure the voltage difference between the two ends of the samples caused by the Seebeck effect. Based on the measurement principle of TEP, the apparatus was built according to the diagram shown in Figure 2. The sample size was 70 × 5 × 3 mm^3^, and its ends were fixed on the copper cap by bolts. The lower copper cap was regarded as the cold end (T), whose temperature was controlled by heating in ice water to maintain 15 °C. The upper copper cap was regarded as the hot end (T + ΔT), whose temperature was controlled by heating to maintain 25 °C. An Agilent 34975A data collector was used to record the voltage (ΔV) difference at both ends of the sample, and the acquisition step was 5 s. The relative TEP was defined as follows:(1)$$\Delta S=\frac{\Delta V}{\Delta T}$$
where ΔV is the low voltage arising from the Seeback effect between the two ends of the samples.

The Rockwell hardness (HRB) was tested according to the Chinese standard of GB/T 230.1-2018 with a SHBRV-187.5 digital Blackwell & Vickers hardness tester with a margin of error of ±2%. The loading force of the test was 100 Kg, and the duration was 5 s. Five points were tested for each sample, and the mean values were obtained as the final result. For further exploration of the effect of the sequence of cryogenic treatment and aging treatment on the mechanical properties of the composites, the electronic universal testing machine was used for the tensile test. Tensile specimens were prepared in accordance with the GB/T 228-2002 standard; the sample dimension is shown in Figure 3. The strain rate during the tensile process was 0.001/s. Three samples of each treatment (SA20, SCA20, SA20C) were tested and averaged as for a final result.

For the purpose of investigating the change in microstructure of the composite after different treatments, the samples were ground on grit-abrasive papers with a mesh from 400 to 2000. The samples were then mechanically polished, and the polished surfaces were corroded for 10 s with Keller’s reagent. The microstructure and tensile fracture surface were detected using a SU1510 scanning electron microscope (SEM) of, which was made by Hitachi in Japan. In order to further analyze the microstructure of the composites, the specimens were mechanically ground into thin slices with a thickness of 50 μm and then subjected to ion milling for examination by G20 transmission electron microscopy (TEM), which was made by FEI in the USA. The TEM and high-resolution images were Fourier-transformed by Digital Micrograph software, and the diffraction spots were calibrated using CaRIne software. The type of precipitates in the alloy could be determined using this account.

## 3. Results

### 3.1. Hardness and Tensile Properties

The hardness variations of the 15%SiCp/2009Al composites with different aging times are shown in Figure 4. It can be seen that the hardness increases with an extension of aging time. However, the hardness of both SA2 and SCA2 exhibited no obvious change when compared to that of the S and SC, respectively. The hardness of the sample treated with SC was higher than that of the sample treated by S. After aging for 6 h and 10 h, the hardness was obviously increased. With the increase of aging time to 20 h, the improvement in hardness induced by cryogenic treatment became more obvious and was approximately 2–4% higher than that of the samples without cryogenic treatment. In the 15%SiCp/2009Al composites, the Rockwell hardness was affected by the SiC particles and the aluminum matrix. As the SiC-reinforced phase cannot be altered by aging treatment and cryogenic treatment, the change in hardness can be mainly attributed to the microstructure evolution of the aluminum matrix.

In order to investigate the effect of cryogenic treatment and its combination sequence with traditional heat treatment on 15%SiCp/2009Al composites, samples treated with SA20, SA20C, and SCA20 underwent tensile tests. The results are compared and shown in Figure 5 and Figure 6. It can be seen from Figure 5 that the trend of stress–strain curves is consistent in the tensile process for all the specimens. The tensile strength, yield strength, and elongation can be obtained from these curves, as is shown in Figure 6. It can be observed that SA20C shows a slight improvement in tensile strength, while SCA20 demonstrates the opposite effect. However, this change in tensile strength is too small and can be ignored. Both SA20C and SCA20 can improve the yield strength of the 15%SiCp/2009Al composites. The improvement caused by SCA20 is more effective, being 16 MPa higher than that of the SA20 sample. The tendency of reduction in elongation can be observed. Due to the relatively poor plasticity of aluminum matrix composites and the discreteness of elongation, this reduction can also be ignored. Therefore, the most obvious effect caused by cryogenic treatment is the improvement in yield strength when cryogenic treatment is performed after solution and prior to aging treatment.

In order to further reveal the change in yield strength induced by cryogenic treatment, the macro and micro tensile fracture surfaces of SA20 and SCA20 were detected and compared, as is shown in Figure 7 and Figure 8. It can be seen from Figure 7 that the tensile fracture surfaces of both SA20 and SCA20 are flat and have no obvious plastic deformation, which shows that the fracture of the 15%SiCp/2009Al composite is a typical brittle fracture. The micro-fracture surfaces show that the tensile fracture is comprises a majority of brittle dents and a small number of ductile dimples with a small size, as is shown in Figure 8a,d. The inner surface of the dent is relatively smooth, and the shape of the tear edge is consistent with the profile of the SiC particles, marked by the dotted line cycles in Figure 8b,c,f. A large number of studies have shown that there are three main fracture forms of aluminum matrix composites [28,29]: the fracture of reinforcement particles, the decohesion of SiC particles from the matrix, and the fracture of the aluminum matrix. Therefore, the fracture of the samples in both states is dominated by the decohesion of SiC particles from the matrix. There are also a small number of cracks propagating through the particles, which means that the fracture of reinforcement particles exists. Although the macroscopic appearance is that of a brittle fracture, there are a few small-sized dimples in a small range of aluminum alloy matrices, shown in Figure 8e,f, indicating that ductile tearing is the crack propagation mode in aluminum alloy matrices. There are relatively more ductile dimples in the matrix of the sample after cryogenic treatment when compared with that of the sample without cryogenic treatment (see a comparison of Figure 8b with Figure 8e). The tensile strength of the 15%SiCp/2009Al composite is mainly decided by the bond strength between the SiC particles and the aluminum matrix. The plastic deformation behavior of the matrix has a great influence on the yield strength. It can be seen that the variations of interface caused by cryogenic treatment are not enough to cause the change in tensile strength during tensile process. However, the changes in the aluminum matrix induced by cryogenic treatment will be the main reason for the improvement in yield strength.

### 3.2. Microstructure Evolution by SEM

In order to investigate the effect of cryogenic treatment on the precipitation behavior of the 15%SiCp/2009Al composites, the microstructures of the S and SC samples after aging for different times were detected by SEM, shown in Figure 9. It can be seen that the microstructure of the solution-treated (S) 15%SiCp/2009Al composites consisted of SiC particles with a size in the range of 4–10 μm and the recrystallized grain of the aluminum alloy matrix. There was a small amount of precipitates dispersed in the aluminum matrix of the solution-treated sample, appearing as the white particles shown in Figure 9a,d. P. Jin et al. [11] revealed that the remaining coarse particles in the solutionized SiCp/2009Al composites should be undissolved secondary phases, which were identified to be Al_2_Cu (Ɵ phase) and Al_7_Cu_2_Fe phases. The volume fraction of the secondary phases is mainly determined by the solution temperature. The increase in solution temperature can reduce the content of remaining particles. There was no obvious difference in the microstructure between the S and SC samples.

After aging for 6 h (SA6), the white, small-size particles in the aluminum matrix were increased to a small extent when compared with that of the S sample. The aging treatment can promote the precipitation of more secondary particles, which depend on the aging temperature and time [10]. For the same aging time, the addition of cryogenic treatment (SCA6) had the trend of increasing the precipitation phases when compared to that of the SA6 samples, especially for the smaller-size particles, as is shown in Figure 9b,e. After aging for 20 h, the small-size precipitates increased obviously as the white particles shown in Figure 9c,f. With the addition of cryogenic treatment (SCA20), the volume fraction of precipitates appears to be higher than that of the SA20. In the 15%SiCp/2009Al composites, the equilibrium phases generated after aging were usually recognized as the θ (Al_2_Cu) phase and the S (Al_2_CuMg) phase, which depended on the ratio of Cu/Mg [30]. It can be inferred that cryogenic treatment after solution treatment can promote the precipitation of secondary phases. However, the type and precipitation stage of secondary phases induced by cryogenic treatment need further investigation.

### 3.3. ThermoElectric Power Measurement

In order to further confirm the change of precipitation from the global microstructure of the materials, the TEP measurements of S and SC samples after aging treatment with different times were tested and compared. Lattice defects that alter the electrical conductivity or elastic properties of the Al matrix, such as solute atoms or precipitates, have an effect on ΔS [25]. Therefore, the change of solute atoms and precipitates can be characterized sensitively by TEP. The relative TEP value of the as-quenched samples (ΔS0) were taken as a reference, and the change value of ΔS was used to characterize the difference of precipitation and phase in the material; Δ(ΔS) = ΔSt − ΔS0, where ΔSt is the TEP value of the sample after aging at 170 °C for different times. The TEP values of the 15%SiCp/2009 aluminum matrix composites after aging for different times are shown in Figure 10. It can be seen that the TEP curves of the 15%SiCp/2009Al composites can be divided into a descending stage and an ascending stage. With the increase in aging time, the TEP values were first decreased and then increased, and finally tended to be stable for both the S and SC samples. Cryogenic treatment followed with solution (SC) decreased the TEP value apparently when compared with that of the S sample. For the SC samples, the minimum value of the TEP was obtained after aging treatment for 6 h. The TEP of the S sample was also decreased with the increase in the aging time, and the minimum value of the SC sample was obtained after undergoing aging treatment for 10 h.

For an aluminum alloy and its composites, there are two main factors that affect the TEP value during aging treatment [21,25]: (1) with the progress of precipitation, the decrease of solute atoms in the solution can change the value of TEP. The contributions of different solution elements to the TEP of aluminum alloy are varied, as Mg and Cu elements have a positive effect on the TEP while Si, Fe, and Mn elements have a negative effect on the TEP [25]. (2) Second, coherent or semi-coherent precipitation formed during aging has different effects on the TEP, depending on the type, size, and volume fraction of precipitation. In terms of coarse, incoherent precipitation, it is generally believed that it has no effect on the TEP of the alloy. Thus, the change in the TEP during the formation of such precipitation is basically only related to the decrease of solute content in the solution.

During the initial stage of aging, the formation of GP regions and transition phases occurs. The segregation of solute atoms increases lattice defects. Additionally, the precipitation of coherent Al_2_Cu and Al_2_CuMg phases reduces the content of Mg and Cu in the solid solution, which results in the reduction of the TEP. It can be seen that the TEP value of the SC sample was significantly lower than that of the S sample. During the descending stage, the TEP value of the SC sample was also lower than that of the S sample with the extension of aging time. After aging reaches a certain time, the TEP curve began to rise, indicating that the coherent phases had begun to transform into incoherent phases. It can be seen that the minimum values of S and SC are obtained by different aging times. Therefore, the change in aging behavior after cryogenic treatment can be reflected by the change in the TEP value.

### 3.4. Microstructure Detection by TEM

The samples of S and SC aged for 6 h and 20 h were adopted for TEM detection. It can be seen that the dislocation density in the aluminum matrix of the SCA6 sample was higher than that of SA6 sample, shown in Figure 11a,d. Many dislocation lines and some precipitates can be observed in the aluminum alloy and within the grain boundary in Figure 11d. It is well-acknowledged that the addition of SiC particles to the aluminum alloy can increase the dislocation density after quenching [11]. Cryogenic treatment can further increase the dislocation density. During the process of cryogenic treatment, excessive internal stress is generated at the interface due to the large difference of the expansion coefficient between the SiC and the aluminum matrix. The movement of dislocations is easily hindered by SiC particles, resulting in the stacking of dislocations at the interfaces and grain boundaries. The increase of dislocation density can promote the nucleating of precipitates. Some precipitates can be also seen in the grain boundary of SA6, as is shown in Figure 11b. The needle-like precipitates, with a size of about 20 nm, distribute homogeneously in the aluminum matrix in the SCA6 sample, as is shown in Figure 11e. Liu et al. [31] reported that these needle-like morphologies showed that the early-stage microstructure was dominated by GPI; zones and a small fraction of GPII zones. It can be inferred that more uniform nucleating sites can be produced by cryogenic treatment. After aging for 20 h, more precipitates can be generated in the aluminum matrix, interface, and grain boundaries in both SA20 (Figure 11c) and SCA20 (Figure 11f). It seems that the content of precipitates in SCA20 sample is higher than that of the SA20 sample. In order to determine the type of precipitates, a high-resolution transmission electron microscope (HRTEM) and a selected-area electron diffraction (SAED) were used to analyse the crystal structure of the precipitates. It can be seen that the aluminum alloy matrix has a face-centered cubic (fcc) lattice structure, shown in Figure 12c,e. The dark, blocky precipitate within the interface of the SiC and aluminum matrix is the θ′ phase according to the HRTEM and the Fourier transform (FFT) patterns, as shown in Figure 12a,b,d. It has the tetragonal structure with lattice parameters of a = 0.404 nm and c = 0.580 nm [32]. This phase is recognized as the main strengthening phase in Al-Cu alloys [32]. It can also grow into large-sized precipitates with a size of close to 1 μm [31]. Therefore, the white particles in the SEM micrographs are mainly θ′ phase. Precipitates in the grain boundaries are also identified to be θ′ phase according to the results of SAED, as is shown in Figure 13a,b. Therefore, it can be concluded that cryogenic treatment after solution can promote the precipitates of the θ′ phase in the 15%SiCp/2009Al composites.

## 4. Discussion

It can be inferred from the above-mentioned results that cryogenic treatment after solution can obviously facilitate the aging process of 15%SiCp/2009 composites. The resultant higher hardness as well as the lower TEP of the SC sample when compared to that of the S sample can be attributed to the increase in dislocation density and the potential energy difference caused by compressive stress in the aluminum alloy matrix, induced by cryogenic treatment. These two factors are demonstrated by Klug et al. [33] as the main mechanisms for modification to the precipitates following cryogenic treatment. During the process of cryogenic treatment, the lattice shrinkage of the aluminum matrix is hindered by SiC particles, so it is easy to generate dislocation accumulation and improve the dislocation density near the interface. Furthermore, the lattice contraction of an aluminum matrix at cryogenic temperature occurs seriously, while the shrinkage of SiC particles is small. High, internal, residual stress in the interface and even within grains can be generated.

After aging for 2 h, the hardness exhibits no obvious change while the TEP reduction is obvious, meaning that the main change in the microstructure is the formation of GP-zones. There are no obvious equilibrium precipitates formed at this stage. However, the formation of a GP-zone and transitional phase in aluminum alloys can reduce the TEP by both increasing lattice distortion and precipitating Mg and Cu elements. This stage can be considered as the incubation period of aging process. Factors induced by cryogenic treatment followed with solution treatment, such as high dislocation density and internal residual stress, can act as a driving force for the precipitation of an aluminum alloy matrix. Therefore, the hardness of SCA2 is also higher than that of SA2.

After aging for 6 h, the hardness is increased which indicates that obvious precipitates are formed. Meanwhile, the TEP value decreases obviously, which shows that the precipitates in this stage are dominated by the coherent θ″ phase. The lower TEP and higher hardness of SCA6 show that a higher content of the θ″ phase can be induced by the addition of cryogenic treatment. With the extension of aging time, θ″ phases transform into the θ′ phase and θ phase, whose coherent relationships with the matrix are decreased and have little influence over the TEP value. Therefore, the TEP shows a little increase with the reduction of the θ″ phase after aging for longer than 6 h, with regard to the sample with cryogenic treatment. However, the TEP of a solution-treated sample will still keep descending after aging for 6 h, and it begins to rise after aging for 10 h. This indicates that cryogenic treatment increases the aging-precipitation driving force of 15%SiCp/2009 composites. As a result, the transformation of metastable phases in the aging process is advanced. With the extension of aging time, the transformation of θ″ phases into θ′ and θ phases dominates the aging process; therefore, the hardness increases obviously. The higher hardness of the SCA sample compared to that of the SA sample can be attributed to the increase in precipitation content induced by cryogenic treatment. The increase of precipitates enhances its pinning effect on dislocations, as a result of enhancing the yield strength of the 15%SiCp/2009 composites.

## 5. Conclusions

The effects of deep cryogenic treatment on the microstructure and mechanical properties of 15%SiCp/2009 aluminum matrix composites were investigated in this work. The main conclusions are shown below.

(1)TEP measurement can be an effective method for evaluating the precipitation characteristics of 15%SiCp/2009 aluminum matrix composites during aging. Cryogenic treatment after solution promotes the precipitation from the start-stage of aging, which can advance the precipitation time of a stable, non-coherent phase. Cryogenic treatment increases dislocation density, which provides a greater driving force for precipitation during the aging process. It can also promote the precipitation of θ′ phases after aging for 6 h. As a result, the content of the equilibrium precipitates formed after aging at the same time is higher than that of samples without cryogenic treatment.(2)After cryogenic treatment, the hardness of 15%SiCp/2009 aluminum matrix composites is improved. The change in hardness induced by cryogenic treatment becomes more significant with the extension of aging time, which is improved by 4% after artificial aging treatment for 20 h.(3)The cryogenic treatment combined with traditional heat treatment has little effect on the tensile strength and elongation of 15%SiCp/2009 aluminum matrix composites. Cryogenic treatment after solution and aging treatment (SAC) also has little effect on yield strength, while the conduction of cryogenic treatment after solution and prior to aging treatment (SCA) can increase the yield strength by 16 MPa.

Considering the applications of cryogenic treatment in the aerospace and automotive industries, it is more importance to investigate the effect of cryogenic treatment on fatigue properties, wear resistance, corrosion resistance, and the dimensional stability of aluminum matrix composites. Furthermore, as residual stress in aluminum matrix composites is a serious problem, the comprehensive modification of residual stress and mechanical properties has more important value for practical application.

## Figures and Tables

**Figure 1 materials-16-00396-f001:**
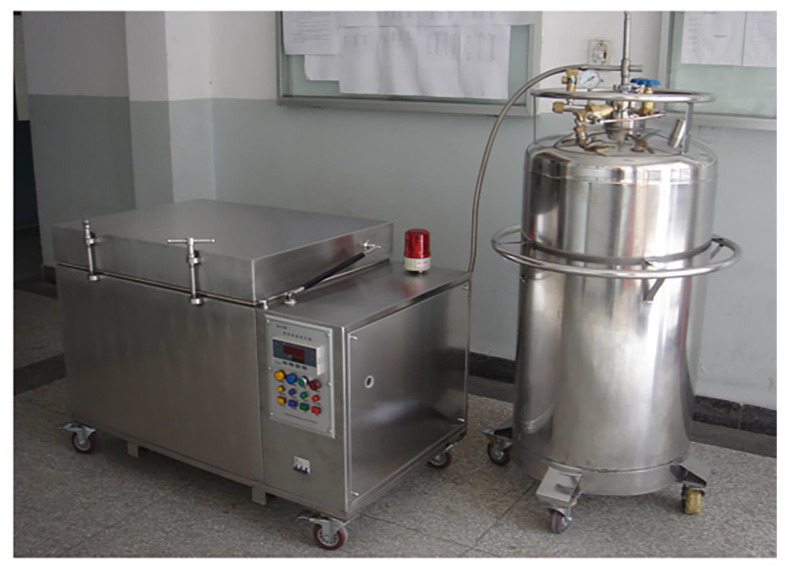
Program-controlled SLX-80 cryogenic system.

**Figure 2 materials-16-00396-f002:**
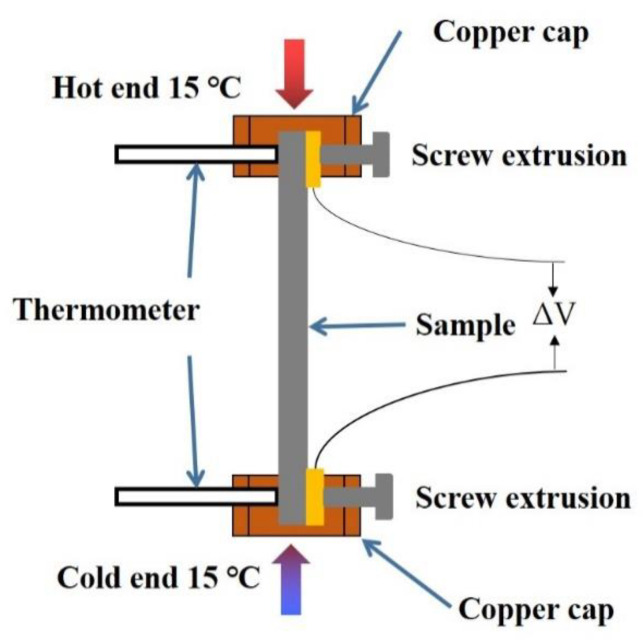
Schematic diagram of TEP measurement.

**Figure 3 materials-16-00396-f003:**
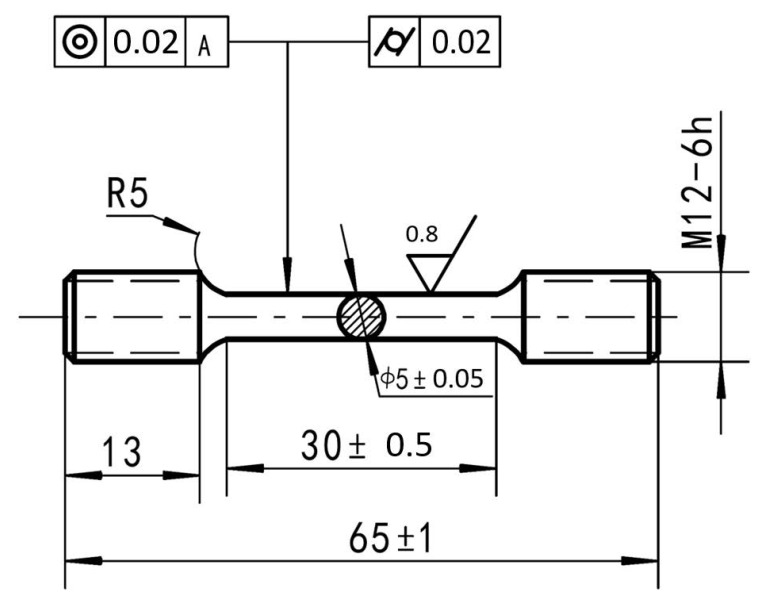
Size and geometry of the tensile test specimens.

**Figure 4 materials-16-00396-f004:**
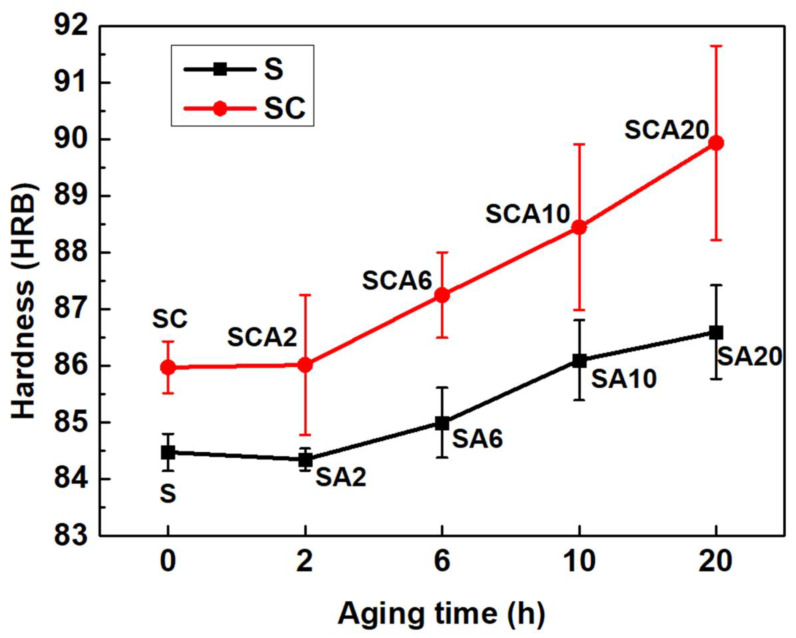
The hardness variations of 15%SiCp/2009Al composites with different aging times.

**Figure 5 materials-16-00396-f005:**
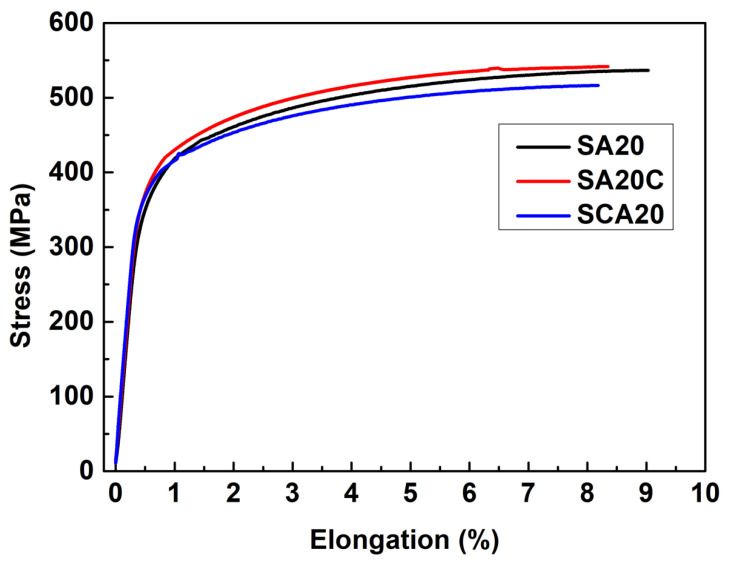
Stress–strain curves of specimens under different treatment conditions.

**Figure 6 materials-16-00396-f006:**
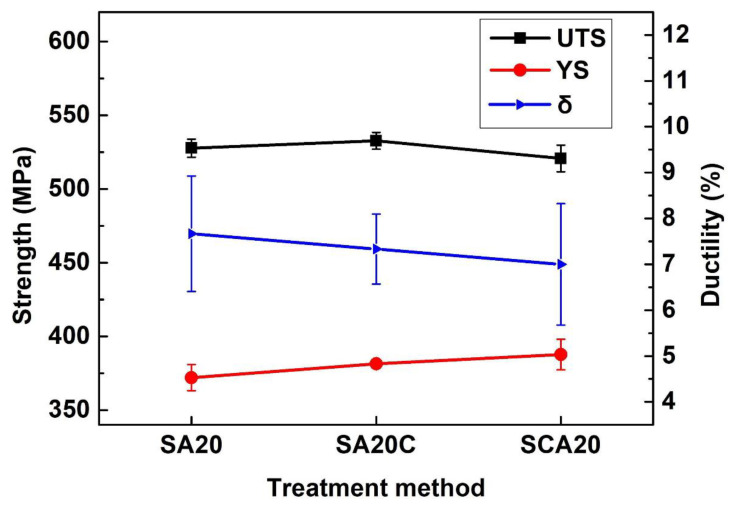
Tensile properties of specimens treated by different processes.

**Figure 7 materials-16-00396-f007:**
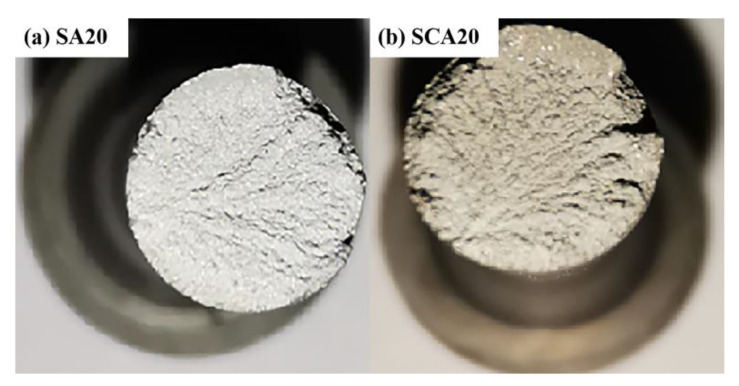
Macro-fracture morphology of the SiCp/2009 aluminum composite, (**a**) SA20 and (**b**) SCA20.

**Figure 8 materials-16-00396-f008:**
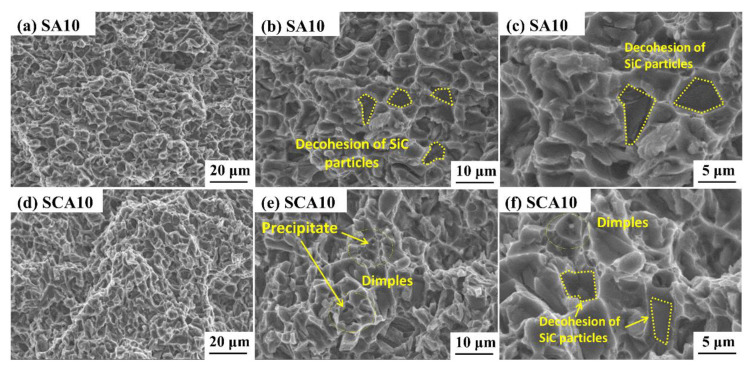
SEM micrographs of tensile fracture surface of samples treated by different processes: (**a**–**c**) without cryogenic treatment; (**d**–**f**) with cryogenic treatment.

**Figure 9 materials-16-00396-f009:**
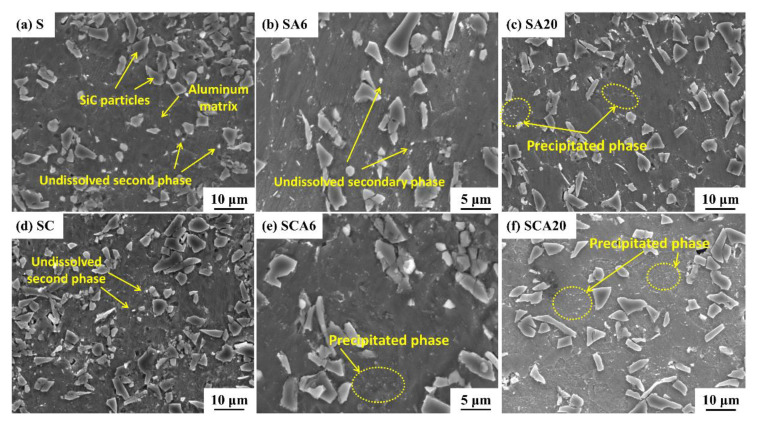
SEM micrographs of SiCp/2009Al composites treated by different processes: (**a**) S; (**b**) SA6; (**c**) SA20; (**d**) SC; (**e**) SCA6; and (**f**) SCA20.

**Figure 10 materials-16-00396-f010:**
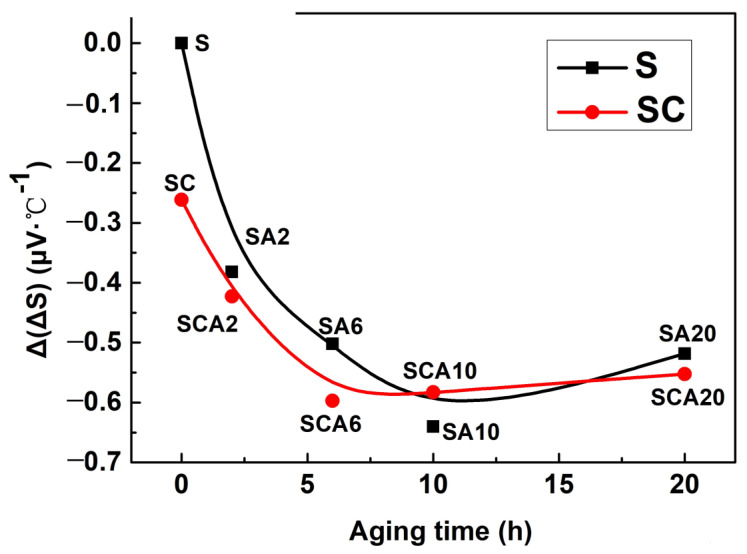
The TEP values of the S and SC samples change with the aging time.

**Figure 11 materials-16-00396-f011:**
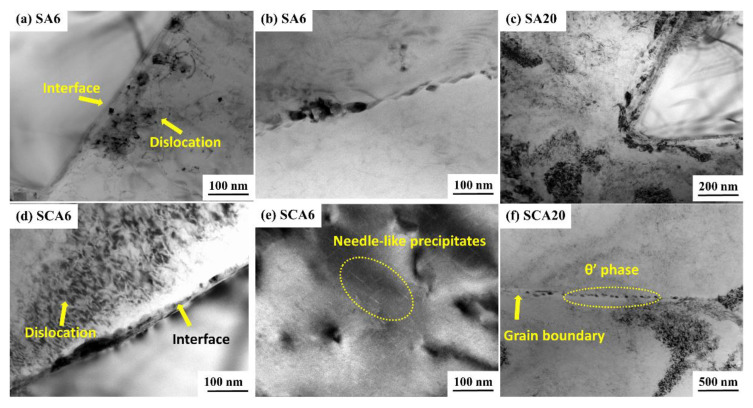
TEM micrographs of the 15%SiCp/2009Al composites treated by different processes: (**a**,**b**) SA6; (**c**) SA20; (**d**,**e**) SCA6; and (**f**) SCA20.

**Figure 12 materials-16-00396-f012:**
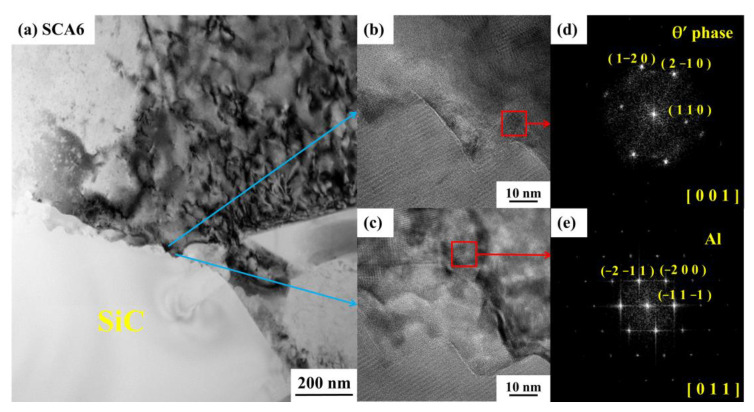
Micrographs of the 15%SiCp/2009Al composites treated by SCA6, (**a**) micrographs of SCA6 sample, (**b**,**c**) HRTEM micrographs of the selected areas in (**a**), (**d**,**e**) the FFT patterns of selected area in (**b**) and (**c**), separately.

**Figure 13 materials-16-00396-f013:**
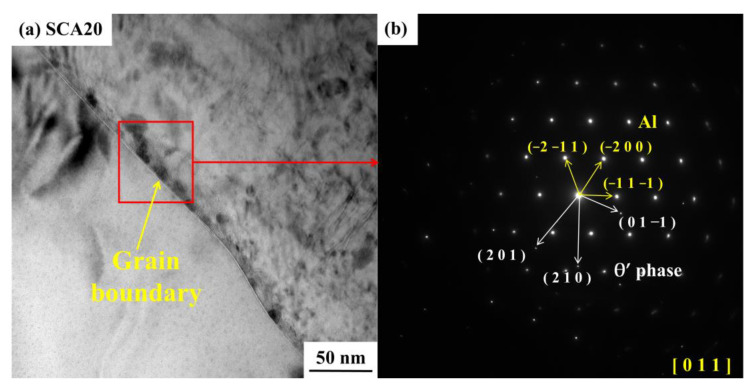
TEM micrographs (**a**) and selected-area electron diffraction (SAED) (**b**) of the 15%SiCp/2009Al composites treated by SCA20.

**Table 1 materials-16-00396-t001:** Chemical compositions of 15%SiCp/2009 aluminum matrix composite (wt%).

Cu	Mg	Si	Fe	Zn	O	Impurity	Al
3.44	1.38	0.29	0.06	0.05	0.15	0.15	Balance

**Table 2 materials-16-00396-t002:** Heat and cryogenic treatment schemes of the 15%SiCp/2009 aluminum matrix composites. The alphabetic order of each sample index indicates the execution sequence of treatments. For example, SCA2 indicates that the sample was treated by solution treatment, cryogenic treatment, and aging treatment for 2 h successively.

Treatment	Sample Index
Solution–Aging treatment	SA2, SA6, SA10, SA20
Solution–Cryogenic–Aging treatment	SCA2, SCA6, SCA10, SCA20
Solution–Aging–Cryogenic treatment	SA20C

## Data Availability

Not applicable.

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
