# Peer review of "The Influence of Cryogenic Treatment on the Microstructure and Mechanical Characteristics of Aluminum Silicon Carbide Matrix Composites"

_materials, 2023, doi:10.3390/ma16010396_

Round 1
Reviewer 1 Report (Previous Reviewer 2)
Desired edits were made by the authors. The article can be published as it is.
comments:
The authors stated that the cryogenic process was performed at -196°C. However, at -196°C, nitrogen turns into liquid form. It creates microcracks on the surface of the material. For this reason, the work done in this area is carried out just below this temperature. If you have done so, please report this temperature exactly. Just because it's a little low doesn't mean that deep cryogenics isn't done. On the contrary, it shows that the operation was done correctly. Otherwise, you must prove that there are no surface cracks.
The authors reported that they applied a 12-hour holding time in the study. The cryogenic process holding time has been the subject of many studies. Authors should at least support the literature on why they chose 12 hours.
It is stated that while the materials are cooled to -196°C, they are cooled at a rate of 1°C per minute. However, it was processed uncontrollably while bringing it to room temperature. Rapid heating may cause microcracks in the material. Doesn't the control of the tank used for the process provide control in bringing it to room temperature? Why was such a method chosen?
Author Response
Dear reviewer,
We deeply appreciate the time and effort you've spent in reviewing our manuscript entitled ‘Effects of cryogenic treatment on the microstructure and mechanical properties of aluminum matrix composites’. Your comments are very helpful and important to us. We have revised the manuscript following the comments exactly. Your comments and our answers are listed as follows.
Comments & Responses
The authors stated that the cryogenic process was performed at -196°C. However, at -196°C, nitrogen turns into liquid form. It creates microcracks on the surface of the material. For this reason, the work done in this area is carried out just below this temperature. If you have done so, please report this temperature exactly. Just because it's a little low doesn't mean that deep cryogenics isn't done. On the contrary, it shows that the operation was done correctly. Otherwise, you must prove that there are no surface cracks.
R: Thank you for your suggestions. We performed cryogenic treatment at -196°C by cooling down samples with a low rate (1°C/min) using program-controlled cryogenic system, and then immersing it in liquid nitrogen to keep the temperature at - 196°C. Thus, the temperature gradient is small during this process, which resulted in small thermal stress. Lower cooling rate can prevent the sample from cracking.
The authors reported that they applied a 12-hour holding time in the study. The cryogenic process holding time has been the subject of many studies. Authors should at least support the literature on why they chose 12 hours.
R: Thank you for your suggestions. In generally, longer holding time is more beneficial to the improvement induced by cryogenic treatment. However, for different alloys and different properties, the optimum holding time may be different. For example, Zhou[1] et al. have reported that 6 hours is the best holding time. Ma[2] et al. have found that 12 hours can obtained the optimum improvement. Li [3] et al. have showed that 36 hours was were the optimum holding time. We will do more work about the effect of holding time in our future work.
[1] Zhou J, Xu S, Huang S, et al. Tensile properties and microstructures of a 2024-T351 aluminum alloy subjected to cryogenic treatment[J]. Metals, 2016, 6(11): 279.
[2] Ma S, Su R, Wang K, et al. Effect of deep cryogenic treatment on wear and corrosion resistance of an Al–Zn–Mg–Cu alloy[J]. Russian Journal of Non-Ferrous Metals, 2021, 62(1): 89-96.
[3] Li C, Cheng N, Chen Z, et al. Deep-cryogenic-treatment-induced phase transformation in the Al-Zn-Mg-Cu alloy[J]. International Journal of Minerals, Metallurgy, and Materials, 2015, 22(1): 68-77.
It is stated that while the materials are cooled to -196°C, they are cooled at a rate of 1°C per minute. However, it was processed uncontrollably while bringing it to room temperature. Rapid heating may cause microcracks in the material. Doesn't the control of the tank used for the process provide control in bringing it to room temperature? Why was such a method chosen?
R: Thank you for your suggestions. The program-controlled cryogenic system can control the heating rate. However, the liquid nitrogen (LN) dewar used for immersion has no this function. So, the usual method is to directly heat samples in the atmosphere after immersion in LN. Due to the slow heating rate of natural convection, microcracks may be avoided. However, the effect of heating rate on the significance of cryogenic treatment is an interesting topic, it is necessary to control both the cooling and heating rate during cryogenic treatment.

Reviewer 2 Report (Previous Reviewer 3)
The authors managed to reply to major concerns that were addressed in my previous evaluating report.
There are only some minor (but important) changes that are required to be considered before (in my opinion) the paper could be considered accepted for publication, like:
1. changing / re-phrasing of the title of this article in order to avoid similarity with other similar papers published in the field (please check the file attached to this report which was generated after it was checked with Ithenticate application) - Instead of "Effects of cryogenic treatment on the microstructure and mechanical properties of aluminum matrix composites" maybe (this is only one suggestion) the title could be "The influence of cryogenic treatment on the microstructure and mechanical characteristics of aluminum silicon carbide matrix composites". This is one recommendation of a new title, but if one better variant of the title (that is not so similar with the one found in the case of other publications) can be found I will not have anything againts it. Important thing is to avoid similarities as much as possible in the title and that the title would be in perfect consent with its contents.
2. rest of similiarities indicated in the file attached to this report - it is just recommended to be slightly reviewed and where it is possible and easy to be rephrased in the paragraphs some of very similar expressions - this could be maybe slightly re-phrased also.
3. There are some minor editing errors in the text - like for instance: "phase phases" (Line 322). One brief reviewing of the paper to do the last spelling and grammar errors that are present in the text is still recommended to be done once more.
4. At the Conclusions sentence - right at the end - it would be great if in one paragraph (few sentences) could be added some ideas / issues to be still addressed in the future (aspects that are foreseen by the authors in their future work) - I have made some suggestions in my previous evaluating report - such as related to the corrosion of the material, fatigue analysis, etc. (aspects that are highly important for the aerospace / automotive domains) and so on! Maybe there are also some important different types of treatments to be considered / other procedures to be used for the evaluating of the mechanical characteristics / other types of alloys to be analyzed in the same manner...
5. In terms of the affiliation of the authors, instead of "a", "b", "c" numbers have to be used - "1", "2", "3" and so on. Postmail addresses of the institutions related to the authors have to be specified as well where affiliations details are being provided in the paper.
Overall, as I said at the beginning, I have seen lot of improvements which were done by the authors. I really appreciate all details, images, new references, ideas, explanations that were added in the text - everything is now much clear.
The remaining issues to be addressed are minor, but important to be considered in my opinion, before the paper could be accepted for publication in the MDPI Materials journal, in the end.

Author Response
Dear reviewer,
We deeply appreciate the time and effort you've spent in reviewing our manuscript entitled ‘Effects of cryogenic treatment on the microstructure and mechanical properties of aluminum matrix composites’. Your comments are very helpful and important to us. We have revised the manuscript following the comments exactly. Your comments and our answers are listed as follows.
Comments & Responses
- changing / re-phrasing of the title of this article in order to avoid similarity with other similar papers published in the field (please check the file attached to this report which was generated after it was checked with Ithenticate application) - Instead of "Effects of cryogenic treatment on the microstructure and mechanical properties of aluminum matrix composites" maybe (this is only one suggestion) the title could be "The influence of cryogenic treatment on the microstructure and mechanical characteristics of aluminum silicon carbide matrix composites". This is one recommendation of a new title, but if one better variant of the title (that is not so similar with the one found in the case of other publications) can be found I will not have anything againts it. Important thing is to avoid similarities as much as possible in the title and that the title would be in perfect consent with its contents.
R: Thank you for your suggestions. We revised the title into “The influence of cryogenic treatment on the microstructure and mechanical characteristics of aluminum silicon carbide matrix composites”
- rest of similiarities indicated in the file attached to this report - it is just recommended to be slightly reviewed and where it is possible and easy to be rephrased in the paragraphs some of very similar expressions - this could be maybe slightly re-phrased also.
R: Thank you for your suggestions. We have revised some of similar expressions in the revised manuscript.
- There are some minor editing errors in the text - like for instance: "phase phases" (Line 322). One brief reviewing of the paper to do the last spelling and grammar errors that are present in the text is still recommended to be done once more.
R: Thank you for your suggestions. We have reviewed the manuscript and checked the spelling can grammar errors.
- At the Conclusions sentence - right at the end - it would be great if in one paragraph (few sentences) could be added some ideas / issues to be still addressed in the future (aspects that are foreseen by the authors in their future work) - I have made some suggestions in my previous evaluating report - such as related to the corrosion of the material, fatigue analysis, etc. (aspects that are highly important for the aerospace / automotive domains) and so on! Maybe there are also some important different types of treatments to be considered / other procedures to be used for the evaluating of the mechanical characteristics / other types of alloys to be analyzed in the same manner...
R: Thank you for your suggestions. We have added a paragraph in the section of conclusions, as show below:
Considering its’ application in aerospace and automotive industries, it is more importance to investigate the effect of cryogenic treatment on fatigue properties, wear resistance, corrosion resistance and dimensional stability of aluminum matrix composite. Furthermore, due to residual stress in aluminum matrix composite is a serious problem, the comprehensive modification of residual stress and mechanical properties has more important value for practical application.
- In terms of the affiliation of the authors, instead of "a", "b", "c" numbers have to be used - "1", "2", "3" and so on. Postmail addresses of the institutions related to the authors have to be specified as well where affiliations details are being provided in the paper.
R: Thank you for your suggestions. We have revised the number of affiliation of the authors and addresses.

Reviewer 3 Report (New Reviewer)
Journal: Materials (ISSN 1996-1944)
Manuscript ID: Materials- 2125623
The authors presented an article on “Effects of cryogenic treatment on the microstructure and mechanical properties of aluminum matrix composites”. The manuscript contains interesting findings and seems well written. It is understood that a comprehensive revision has been made. Considering the introduction, it can be said that the literature review is sufficient for this study. In addition, the experimental section contains detailed information about all the lines of the study. In the Results section, many analyzes such as microstructure, mechanical properties and Thermo Electric Power measurement of SiCp/2009Al composite were made and remarkable results were obtained. In addition, the observed results are well discussed with the articles published in the literature. In my opinion, this article can be accepted for publication in the "Materials" journal in its final form.

Author Response
Dear reviewer,
We deeply appreciate the time and effort you've spent in reviewing our manuscript entitled ‘Effects of cryogenic treatment on the microstructure and mechanical properties of aluminum matrix composites’. Your comments are very helpful and important to us. We have revised the manuscript following the comments exactly. Your comments and our answers are listed as follows.
Comments & Responses
- The authors presented an article on “Effects of cryogenic treatment on the microstructure and mechanical properties of aluminum matrix composites”. The manuscript contains interesting findings and seems well written. It is understood that a comprehensive revision has been made. Considering the introduction, it can be said that the literature review is sufficient for this study. In addition, the experimental section contains detailed information about all the lines of the study. In the Results section, many analyzes such as microstructure, mechanical properties and Thermo Electric Power measurement of SiCp/2009Al composite were made and remarkable results were obtained. In addition, the observed results are well discussed with the articles published in the literature. In my opinion, this article can be accepted for publication in the "Materials" journal in its final form.
R: Thank you for your suggestions.

Reviewer 4 Report (Previous Reviewer 1)
The authors have successfully addressed all the comments and concerns, that is why the manuscript is suggested to be accepted in the present form.
Author Response
Dear reviewer,
We deeply appreciate the time and effort you've spent in reviewing our manuscript entitled ‘Effects of cryogenic treatment on the microstructure and mechanical properties of aluminum matrix composites’. Your comments are very helpful and important to us. We have revised the manuscript following the comments exactly. Your comments and our answers are listed as follows.
Comments & Responses
- The authors have successfully addressed all the comments and concerns, that is why the manuscript is suggested to be accepted in the present form.
R: Thank you for your suggestions.

Reviewer 5 Report (Previous Reviewer 4)
The authors have addressed all of the previous comments and have substantially improved the manuscript. Through the reading of the manuscript I could only find some typos or formatting errors:
Line 68: I think it should be β’ instead of β’’
Figure 8: in (e) the notation "precipitate" is written without capitalization of the first letter, whereas everywhere else the first word is capitalized. Also the "Plastic dimples" can be written as simply "Dimples" as they are inherently of plastic nature.
Figure 9 and 11: Same thing as for Figure 8, some words are capitalized and others not.
Author Response
Dear reviewer,
We deeply appreciate the time and effort you've spent in reviewing our manuscript entitled ‘Effects of cryogenic treatment on the microstructure and mechanical properties of aluminum matrix composites’. Your comments are very helpful and important to us. We have revised the manuscript following the comments exactly. Your comments and our answers are listed as follows.
Comments & Responses
The authors have addressed all of the previous comments and have substantially improved the manuscript. Through the reading of the manuscript I could only find some typos or formatting errors:
- Line 68: I think it should be β’ instead of β’’
R: Thank you for your suggestions. We have reviewed the reference again, the β’’ in line 69 should be β’, we have revised it.
- Figure 8: in (e) the notation "precipitate" is written without capitalization of the first letter, whereas everywhere else the first word is capitalized. Also the "Plastic dimples" can be written as simply "Dimples" as they are inherently of plastic nature.
R: Thank you for your suggestions. We have revised the words in Figure 8.
- Figure 9 and 11: Same thing as for Figure 8, some words are capitalized and others not.
R: Thank you for your suggestions. We have revised the words in Figure 9 and 11.

This manuscript is a resubmission of an earlier submission. The following is a list of the peer review reports and author responses from that submission.
Round 1
Reviewer 1 Report
The article that has been submitted to the selected journal has an emphasis cryogenic treatment of Al-alloy. Some general remarks regarding the topic: the topic has become after a longer hiatus in this research field more interesting for research in the last five years, due to the pressure of general public and science as a response to the climate impact, natural resources and development of various industries (medicine, robotics, automotive industry, 3D printing, aerospace etc.). The topic is of high interest for the selected journal; however, the submitted manuscript looks more as a technical report than an article. In the current state the manuscript is suggested for major review, if the comments/remarks are addressed accordingly:
-The manuscript in not well written, in regards to scientific soundness, inappropriate designations/citation/reference style, etc., figures are not well presented or marked etc.
-There is a high lack of discussion and evaluation of the results and comparison to other literature (literature review is insufficient (there is lack of references in from 2018 onwards, where new understanding and discoveries were published in correspond to cryogenic treatment in regards to Al-alloy). The literature is highly biased towards certain authors/groups, without reporting other groups, which have also done research in this field! In order to avoid this, the following groups (here are presented just some of them) have to be checked and citated accordingly! The groups are: Steier et. al., Koklu et al.: Jovicevic-Klug et al., Gao et al., Park et al., Mei et al. and Krymsky et al. There is also a group in Toulose, France, which also works on CT of Al-alloys, check this group also!
-In regards to introduction, what kind of cryogenic treatments are known? Please add accordingly.
-The aims of the article have to be rewrite in order to really emphases how this research is different from previous ones and what is the new contribution.
-In introduction is mentioned -108 C (shallow cryogenic treatment), whereas in methods it is used deep cryogenic treatment (-196 C). This is not consistent and with different cryogenic temperature come also different CT mechanisms and microstructure transformation. Find the new refence(s) or make a new experiment.
-Why did you choose only 12 h for DCT of Al-alloy, as it has been showed the most optimal treatment is exposure for at least 24 h, for non-ferrous even 48h!
-Please define according to which standard did you measure hardness.
-Why did you use Keller´s reagent for etching?
-Please add/mark the groups/tests on the graphs, so that the reader can recognize for each group is the description. As in current state, is very confusing.
-In regards to microstructure, the microstructure analysis has to be rewritten in order to improve the message and the most important part of the manuscript, as such microstructure influence on the final properties of the alloy! Figure 4 and figure 5 have to be improved in current state the quality of the images is bad and the marked features, especially on figure 4 are not seen or are highly doubtful. Also mark all the features on the given figures. And also, on figure 4 are observed also precipitates, which are not marked!
-Additional comments to figure 5:-is this really grain boundary or it is just an effect of the etching? Again, if it is, please improve the quality of the image and also include higher magnification images to confirm this.
-Add in the microstructure part TEM results.
-Comment to figure 7 (magnify the scale bar) and results correlated to this: due to presence of Mg, O and Si, the particles which are observed could be dispersoids MgO and MgSi2, which are formed within the given alloy. Example can be found in Schuster et al. 2021 (https://doi.org/10.1016/j.matdes.2021.110131). What is marked as dislocation can be anything, this is not a proper proof. Moreover, point EDS made on 50 nm scale is not a proper technique to identifying the selected phase, as such the surroundings is also included in this analysis.
-Discussion and conclusions part have to be improved and updated accordingly to new understanding of the microstructure.
Author Response
Dear reviewer,
We deeply appreciate the time and effort you've spent in reviewing our manuscript entitled ‘Effect of cryogenic treatment on the ageing behavior of 15% SiCp/2009 aluminum matrix composites’. Your comments are very helpful and important to us. We have revised the manuscript following the comments exactly. Your comments and our answers are listed as follows.
Reviewer 1:
The article that has been submitted to the selected journal has an emphasis cryogenic treatment of Al-alloy. Some general remarks regarding the topic: the topic has become after a longer hiatus in this research field more interesting for research in the last five years, due to the pressure of general public and science as a response to the climate impact, natural resources and development of various industries (medicine, robotics, automotive industry, 3D printing, aerospace etc.). The topic is of high interest for the selected journal; however, the submitted manuscript looks more as a technical report than an article. In the current state the manuscript is suggested for major review, if the comments/remarks are addressed accordingly:
- The manuscript in not well written, in regards to scientific soundness, inappropriate designations/citation/reference style, etc., figures are not well presented or marked etc.
Thank you for your suggestions, we have revised the manuscript according to your comments carefully.
- There is a high lack of discussion and evaluation of the results and comparison to other literature (literature review is insufficient (there is lack of references in from 2018 onwards, where new understanding and discoveries were published in correspond to cryogenic treatment in regards to Al-alloy). The literature is highly biased towards certain authors/groups, without reporting other groups, which have also done research in this field! In order to avoid this, the following groups (here are presented just some of them) have to be checked and citated accordingly! The groups are: Steier et. al., Koklu et al.: Jovicevic-Klug et al., Gao et al., Park et al., Mei et al. and Krymsky et al. There is also a group in Toulose, France, which also works on CT of Al-alloys, check this group also!
Thank you for your suggestions, we have rewritten the part of introduction and added the discussion and evaluation of literatures about cryogenic treatment. The works of all the authors and groups your mentioned were quoted in the manuscript.
- In regards to introduction, what kind of cryogenic treatments are known? Please add accordingly.
Cryogenic treatments are usually divided into shallow cryogenic treatment (-80℃) and deep cryogenic treatment (-196℃) according to the minimum cryogenic temperature. In generally, deep cryogenic treatment with lower temperature can obtain better improvement effect.
- The aims of the article have to be rewrite in order to really emphases how this research is different from previous ones and what is the new contribution.
Thank you for your suggestions, the aims of the article have been rewritten as below:
Therefore, the present work aims to investigate the change of aging behavior induced by the conducting of cryogenic treatment after solution treatment of 15%SiCp/2009Al aluminum matrix composites. The methods of thermoelectric power (TEP) and microhardness are used for evaluating the microstructure evolution after aging for different time. As a result, the changes of different aging stage caused by cryogenic treatment can be revealed. Moreover, in order to explore the effect of cryogenic treatment sequence combined with solution and aging, the change of tensile properties induced by cryogenic treatment before aging and after aging are also tested, respectively.
- In introduction is mentioned -108 C (shallow cryogenic treatment), whereas in methods it is used deep cryogenic treatment (-196 C). This is not consistent and with different cryogenic temperature come also different CT mechanisms and microstructure transformation. Find the new refence(s) or make a new experiment.
Thank you for your suggestions. As different materials have different changes of microstructure caused by cryogenic treatment, and the optimum temperature is not the same. In generally, deep cryogenic treatment (-196℃) is more effective than that of shallow cryogenic treatment. Lower cryogenic temperature can obtain greater improvement.
- Why did you choose only 12 h for DCT of Al-alloy, as it has been showed the most optimal treatment is exposure for at least 24 h, for non-ferrous even 48h!
Thank you for your suggestions. In generally, longer soaking time at minimum temperature of cryogenic treatment can obtain better effects. However, the increase of improvement after a certain long time is so small. Therefore, holding at - 196℃ for 12 h has been able to make the specimen temperature uniform and produce relatively obvious changes on the microstructure. In the subsequent research work, it may be a good idea to continue to explore the best holding time of cryogenic treatment. The effects of different holding time and even other parameters such as cooling/heating rate and temperature in cryogenic treatment would be further investigated in our future work.
- Please define according to which standard did you measure hardness.
Thank you for your suggestions. The Rockwell hardness (HRB) were test according to the Chinese standard of GB/T 230.1-2018. We have added this content in the part of Materials and Methods.
- Why did you use Keller´s reagent for etching?
The Keller´s reagent is suitable for etching of aluminum alloy and aluminum matrix composites. And many references also adopted the Keller´s reagent for etching of aluminum alloy and aluminum matrix composites.
- Please add/mark the groups/tests on the graphs, so that the reader can recognize for each group is the description. As in current state, is very confusing.
Thank you for your suggestions. We have added the group number in the graphs.
- In regards to microstructure, the microstructure analysis has to be rewritten in order to improve the message and the most important part of the manuscript, as such microstructure influence on the final properties of the alloy! Figure 4 and figure 5 have to be improved in current state the quality of the images is bad and the marked features, especially on figure 4 are not seen or are highly doubtful. Also mark all the features on the given figures. And also, on figure 4 are observed also precipitates, which are not marked!
Thank you for your suggestions. We have rewritten the microstructure analysis and revised Figure 4 and figure 5, as shown in the manuscript.
- Additional comments to figure 5:-is this really grain boundary or it is just an effect of the etching? Again, if it is, please improve the quality of the image and also include higher magnification images to confirm this.
We feel sorry that the micrographs and marks are not very clearly. We have revised the graphs and changed the marks.
- Add in the microstructure part TEM results.
We have added micrographs of TEM, HRTEM and SAED, and also revised the content of TEM results in the manuscript.
- Comment to figure 7 (magnify the scale bar) and results correlated to this: due to presence of Mg, O and Si, the particles which are observed could be dispersoids MgO and MgSi2, which are formed within the given alloy. Example can be found in Schuster et al. 2021 (https://doi.org/10.1016/j.matdes.2021.110131). What is marked as dislocation can be anything, this is not a proper proof. Moreover, point EDS made on 50 nm scale is not a proper technique to identifying the selected phase, as such the surroundings is also included in this analysis.
Thank you for your suggestions. We have magnified the scale bar. Studies showed that the type of precipitates in 15%SiCp/2009Al composites are mainly Al2Cu (θ' phase) and Al2CuMg (S') depending on the ratio of Cu/Mg. We have added the HRTEM and SAED of some precipitates, and compared the crystal structure, lattice parameters and the morphology with that in the reference. As a result, we found that precipitates in the interfaces and grain boundaries are mainly dominated by θ' phase. We have changed the TEM micrographs refers to the dislocation and also deleted the results of EDS.
- Discussion and conclusions part have to be improved and updated accordingly to new understanding of the microstructure.
We have improved and updated the discussion and conclusion.

Reviewer 2 Report
Manuscript Number: Materials-2031814
Title: Effect of cryogenic treatment on the ageing behavior of 15% 2SiCp/2009 aluminum matrix composites
Decision: Minor revision
Article Type: Article
The paper is not acceptable for publication in its present form. The authors may revise the article considering the following comments.
Ø The abstract section is the most important part that determines whether the article will be read by the reader. For this reason, the results and the obtained ratios should be given in a way that attracts attention.
Ø Although the cryogenic process has an important place in the study, very little is given in the introduction. In the introduction, the effects of cryogenic treatment on the metallurgical properties of materials can be discussed in more depth. These studies should not be limited to composite materials or aluminum materials.
Ø Since nitrogen is in liquid form at -196°C, the cryogenic process is performed by immersing it in the liquid when performed at this temperature. This process also causes microcracks on the material. Was -196°C seen in the cryogenic treatment tank in the study? If it was seen, it was not stated that the experiments were done by immersing in liquid nitrogen. If it is not seen, the temperature at which it was made at the lowest temperature should be specified. If it is at a temperature lower than -125°C, it will be deemed to have already been subjected to deep cryogenic processing. (Deep and shallow cryogenic process temperatures can be given in the introduction). The image of the cryogenic treatment tank should be shared.
Ø It is said that with 1°C per minute, the material drops to -196°C. That's a decent rate for the cryogenic process. However, bringing the material to room temperature after cryogenic treatment provides uncontrolled cooling. This is an undesirable test method. However, if it is not possible, it is acceptable. However, the time to reach room temperature should be reported.
Ø Let's add the time for the material to come to room temperature after aging.
Ø There are many studies on the effects of the holding time and holding temperature of the cryogenic process. The authors should explain together with the literature why they chose the claimed -196°C and why they kept it for 12 hours.
Ø Please provide the sampling method and geometry of tensile specimens.
Ø Please provide more detailed tensile results, including tensile curves and fracture diagram, and explain why?
Ø Please give macro and micro images after the tensile test.
After making the above corrections would recommend this article for publication in Materials.
Author Response
Dear reviewer,
We deeply appreciate the time and effort you've spent in reviewing our manuscript entitled ‘Effect of cryogenic treatment on the ageing behavior of 15% SiCp/2009 aluminum matrix composites’. Your comments are very helpful and important to us. We have revised the manuscript following the comments exactly. Your comments and our answers are listed as follows.
Reviewer 2:
- The abstract section is the most important part that determines whether the article will be read by the reader. For this reason, the results and the obtained ratios should be given in a way that attracts attention.
Thank you for your suggestions, we have revised the abstract according to your comments carefully.
- Although the cryogenic process has an important place in the study, very little is given in the introduction. In the introduction, the effects of cryogenic treatment on the metallurgical properties of materials can be discussed in more depth. These studies should not be limited to composite materials or aluminum materials.
Thank you for your suggestions, we have added the content about the research progress of cryogenic treatment in the introduction.
- Since nitrogen is in liquid form at -196°C, the cryogenic process is performed by immersing it in the liquid when performed at this temperature. This process also causes microcracks on the material. Was -196°C seen in the cryogenic treatment tank in the study? If it was seen, it was not stated that the experiments were done by immersing in liquid nitrogen. If it is not seen, the temperature at which it was made at the lowest temperature should be specified. If it is at a temperature lower than -125°C, it will be deemed to have already been subjected to deep cryogenic processing. (Deep and shallow cryogenic process temperatures can be given in the introduction). The image of the cryogenic treatment tank should be shared.
Thank you for your suggestions. For cryogenic treatment (C), a cooling rate of 1 ℃/min is adopted in this work to cool down to - 196℃in order to avoid microcracks in the material. The program-controlled SLX-80 cryogenic system was used for carrying out cryogenic treatment. The image of the cryogenic treatment tank and deep cryogenic process temperature has been added to the manuscript.
- It is said that with 1°C per minute, the material drops to -196°C. That's a decent rate for the cryogenic process. However, bringing the material to room temperature after cryogenic treatment provides uncontrolled cooling. This is an undesirable test method. However, if it is not possible, it is acceptable. However, the time to reach room temperature should be reported.
Thank you for your suggestions, we brought out the material and put it directly into the environment to warm up after cryogenic treatment. The time to reach room temperature is about 15 minutes due to the sample is small.
- Let's add the time for the material to come to room temperature after aging.
Thank you for your suggestions, we have added the time for the materials to come to room temperature after aging, which was about 10 minutes.
- There are many studies on the effects of the holding time and holding temperature of the cryogenic process. The authors should explain together with the literature why they chose the claimed -196°C and why they kept it for 12 hours.
Thank you for your suggestions. As different materials have different changes of microstructure caused by cryogenic treatment, and the optimum temperature is not the same. In generally, deep cryogenic treatment (-196℃) is more effective than that of shallow cryogenic treatment. Lower cryogenic temperature can obtain greater improvement. In generally, longer soaking time at minimum temperature of cryogenic treatment can obtain better effects. However, the increase of improvement after a certain long time is so small. Therefore, holding at - 196℃ for 12 h has been able to make the specimen temperature uniform and produce relatively obvious changes on the microstructure. In the subsequent research work, it may be a good idea to continue to explore the best holding time of cryogenic treatment. The effects of different holding time and even other parameters such as cooling/heating rate and temperature in cryogenic treatment would be further investigated in our future work.
- Please provide the sampling method and geometry of tensile specimens.
Thank you for your suggestions, we have provided the sampling method and geometry of tensile specimens
- Please provide more detailed tensile results, including tensile curves and fracture diagram, and explain why?
Thank you for your suggestions, we have added the tensile curves and macro surfaces of tensile tests. The fracture characteristics were explained in the manuscript.
- Please give macro and micro images after the tensile test.
Thank you for your suggestions, we have added the macro and micro images of the fracture surface both together.
- After making the above corrections would recommend this article for publication in Materials.
Thank you for your recommendation.

Reviewer 3 Report
After a thorough analysis of the paper prepared by the authors, I can appreciate that they are struggling to improve it!
1. In my opinion from the scientific point of view, I don't find this paper as being very innovative. The authors fails to emphasize what this papers brings new in comparison with the other similar researches that were done by other researchers in this domain.
There are references to other works done by other researchers in this domain, but what was the reason of this paper - to do a repetiton of the research done by the others? Is there anything innovative in the way the samples were manufactured or prepared for the tests? It is something new in the recipe of the material as compared with the one done by the others (just by showing chemical composition, I didn't get if there is something new in the recipe or not)?
Using of ThermoElectric Power measurement is a new innovative method that the researchers used for the evaluation or it is something that was done different as compared to the way the other are applying this method for the evaluation?
SEM, TEM analyses are not very spectacular! Why there are not included images taken from TEM microscope when chemical ratio it is determined?
Why EDS and not EDX? Why not XRD analyses to be used in this case?
Why tensile strength was considered as being important in this case and not other mechanical characteristics like compression or fatigue?
Why there were not taken into consideration aspects related to the corrosion of the material?
Why hardness is so critically important in this case when aluminum matrix is used?
Are there any standards to be reffered to for the aerospace and automotive domains? It is petty that just in one sentence these domains are being reffered to!
References related to this paper are quite old (only one reference - number 22 is dated on 2020, the rest are from 5 years ago (most of them) or even older / out dated!!!
This is not at all a very good feeling when I am seing this!
I was expecting that out of 26 sources...15 will be dated on 2021 or 2022...
2. There are also lot of issues regarding the writting of this paper....Technical terms that are inconsequent - for instance there are specified (in the conclusions section) details about macro hardness (like in the paper someone presented details also about microhardness, which obviously it is not the case)...in all cases are presented details in the preamble about "hardness" or "harness" (Line 313). Please notice that there are errors from the English spelling and grammar point of view!
References to the figures are incomplete! For instance in case of Fig. 4 are presented several images like for (a) to (f) but in text all are referenced in a single phrase as Fig.4. The same issue occurs in the case of the other figures...like for instance in case of Fig. 5 there are presented images from (a) to (f) also but in the text explanations are provided just for the images (b), (d) and (e). Why there are not given at all explanations related to the images (a), (c) and (f) in this case? Same trend can be seen also in the other cases / other figures in general!
Also in terms of affiliations of the author there are references like "a", "b" and "c" in the text instead of "1", "2" and "3"! How so?
Even the title "Effect of cryogenic treatment on the ageing behavior of 15% 2 SiCp/2009 aluminum matrix composite" how it is formulated is not correlated with the content of the paper / tests and results reached in the paper at all in my opinion.
Taking into consideration all these, my personal feeling is that this paper in this current form is a borderline paper. I definitely not recommend the paper to be published in the Materials MDPI journal in this form - I am more close to rejecting it instead of accepting it, but I will finally go with the decision of reconsider it after major (but please consider it major major major / extra major revision). Lot of improvements needs to be done before the paper could be considered for publication in MDPI Materials journal.
Reviewer 4 Report
Lines 38-40 needs reference
Bulk reference in line 38 should be segmented and the individual references associated with the individual properties mentioned in the sentence.
Line 38, which aluminum alloy? It is never mentioned that it is 2xxx or other. From the references it is clear we are not only discussing the 2xxx series. Authors should make this clearer.
In line 43 the authors should specify the chemical composition (can be set as the most generic one) for the general reader to follow what the GP, S’’, S’ and S phases are.
Lines 44-45 the authors chose an old reference about SIC enhancement of aluminum alloys, but why did they choose this one? The alloy in the references is a 7xxx series, which has completely different precipitation dynamics than the 2xxx series. Furthermore, the reference is relatively superficial on the effect of the dislocation density and stress effect on the precipitation dynamics. There are many new articles from the last 5 years that more clearly discuss this phenomenon.
Line 55-58: The authors state the heat treatment is done to enhance the bonding strength of the matrix with the dispersion particles. The authors should avoid writing such fallacies. The SiC particles are strongly incoherent particles and will not bond directly with the matrix. Also, they are high temperature resistant particles (check a phase diagram) and it is clear that the particles cannot be heat treated with the low temperature treatments common for aluminum alloys. The heat treatments (both solution and aging) are done to enhance the matrix (presetting and inducing precipitation hardening) and nothing else. Such superficial explanations of indicating a higher bonding energy are not correct and only deepens the misunderstanding of the effect of heat treatment on aluminum composite materials.
The explanation as discussed based on the references 17-18 should be revised. The authors misinterpreted what bonding strength is discussed and which impact comes from the aluminum matrix strengthening, what comes from the dispersion hardening, what comes from the dislocation impaction and recombination and what comes from the precipitation hardening. There should be much more discussion on this and a deeper review of this topic is needed. It is clear that the authors did not conduct a thorough literature survey, which lead to the strong misinterpretations that also follow later in the text.
Lines 67-77: The selected references are outdated and randomly selected. The discussion about the aging kinetics and influence of inclusions, dispersoids and pre-aging nuclei has developed significantly since then. The authors should spend more time in doing a proper literature review on this topic as there is no shortage of much better and deeper articles on this topic. Also the authors should either stick to one al alloys system, namely the 2xxx series or do a broader literature review. Either way the authors need to strongly modify the introduction part on the general heat treatment of targeted aluminum alloys and their composites. An important reference should be the paper from Araghchi et al. and similar.
Line 72: The author name of reference 21 is written all in capital letters.
Lines 82-85: The information presented by the references in this sentence is not helpful as it only gives a superficial indication on this phenomenon. The paper from Bouzada et al. and two papers from Jovicevic-Klug et al. give a deeper insight into the precipitation dynamics with cryogenic treatment on a fundamental level with a deep insight into the microstructural effect of cryogenic treatment.
Line 103-106: The reference 26 should be added here.
Line 149: The authors use new sample designations Sa20, SCA20 and SA20C. They should be explained and added in Table 2.
Line 168: Did the authors measure hardness with Rockwell method? Why not the Brinell hardness, this would be much more accurate and reliable than the HRB for this material.
Line 169-171: What about effect of pre-strained material and residual stresses? How can the authors be sure of the precipitation reasoning, if they have no concrete evidence to support this? The authors should also consider the effect of cryogenic contraction of the matrix and differences in the thermal expansion. How is the final hardness in the case of SCA and SA after 20 h of aging? Why do the authors only mention the 2 h aged case?
Figure 3: The authors state that there is yield strength difference. Since the authors claim there is a difference in the yielding capabilities, which should relate to the cohesion of the matrix with SiC, then also the shape of the tensile curve should change as the strain energy evolution is proportional to the bonding alignment. If this is the case the authors should see a shift in the energy stored in the material before breaking. If not, then this is not the reason for the increased yield. The authors should present and discuss the tensile curves to support their claims.
The lines 210-213 already go into the previously mentioned topic. More understanding is needed about the tensile curves!
The authors claim there is more ductile fractures visible for the SC sample. The authors do not present anything convincing or giving any reliable values in terms of numbers or fractions to support this. The fractography seems relatively meaningless and is left more to the imagination of the authors than to any proven scientific research and explanation.
Line 223-224: May is a word that indicates that the authors did not conduct enough investigation to understand what is in the material. It either is or is not. If the authors do not know from where the precipitates originate, then the research of later processing is meaningless.
Section 3.2: The authors present some general SEM images with very low magnification. The authors should spend more time and effort in analyzing the supposed precipitated particles. They should evaluate their morphology and distribution. Do chemical Analysis of the particles with EDS mapping or EDS point analysis. The precipitates can be relatively well analyzed based no the size seen from the current Sem images (they are currently at a magnification like what a conventional optical microscope can deliver, so not sure why the authors chose such images. In its current state the SEM analysis is meaningless as it does not provide any more information nor couple well together with the other results. Also how do the authors sometimes mark intragranular precipitation and sometimes the grain boundary precipitation. They do not disclose much information about this nor the differences between the precipitates and their formation in individual samples.
Section 3.3: The authors discuss that the TEP is directly proportional to the formation of the precipitates, but then do not clearly explain why the TEP rises faster for the SC than for S, but the SC never boomerangs above the value of S. This should be more clearly discussed and explained as it is important for the fundamental connection of the TEP to the precipitation dynamics.
Section 3.4: The authors discuss that there are more dislocations for the SCA6 than for SA6, but the regions are small and arbitrary and the dislocations are pilled-up on a random position. The authors should spend more time and effort in evaluating in a more statistically relevant fashion. Either through several TEM imaging positions with more clear randomized dislocation ensemble or using ECCI to analyze the dislocation density using SEM.
The authors should provide a deeper investigation with TEM, namely the SAED of the individual precipitates and phases. The EDS analysis of the regions with the needles is very skeptical as this would imply that the precipitates are the only carriers of the alloying elements and the matrix should be already released of all alloying elements. This cannot be as then there would be no more aging effect, which is seen to not be the case. This basically means that the chemical composition of the proposed precipitates is strictly wrong based on the derived EDS data. Also why did the authors disregard the Si effect on the precipitates? The Si is very commonly bonded also within the precipitates as seen throughout all the aluminum alloy systems. Also, to this regard, what is the EDS from the SA6 sample? How does this evolve in the SCA20 and SA20 samples? All in all, the authors should provide more evidence that it is a certain precipitate and check with TEM alignment and zone axes that they are not simply “missing” the precipitated phases because of different viewing plane.
Line 302: The authors should do a thorough investigation and not simply state it may be this or that. Science should be done correctly and not arbitrarily assuming something without proof and clear-cut results. Generally. The TEM images show that there are precipitates, but based on the scale of them they are not connected with the precipitates from the SEM images. As such the intermediately scaled precipitates are not evaluated at all, leading to a major missing understanding of the microstructural evolution.
In the discussion part:
Lines 315-318: The authors consider the stress changes etc. but never measured this or confirmed this. As such this part of the discussion needs to be supported by data not by assumptions.
Line: 318-320: The authors claim that TEP after 2h is associated to the GP zones formation. However, no proof is provided on this. The authors did not present any data from 2h about the GP zones formation nor giving any additional result to confirm the TEP evaluation. To this end the conclusion of the TEP data is highly dubious.
Lines 327-333: How did the authors confirm that the corresponding TEP values and their change with time is directly related to these precipitates. Why other precipitates could not be the reason? How can this be related to different precipitation activation in more chemically rich or depleted zones. How is it related to stress relaxation with aging? It seems the authors simply specify one option without giving a clear indication about the other possible influencing parameters or effects. All in all the conclusions provide no clear mechanism or role of cryogenic treatment on the precipitation other than saying that it is influences as we change in hardness and change in TEP, but no clear indication what changed and how.
With the above comments and questions, I feel the manuscript is lacking in understanding of the topic and literature survey. To this end the authors also provide a really basic introduction and state of the art and it feels that the authors had a clear referencing bias. With that also the results do not provide meaningful conclusions and do not support the proposed conclusions and suggestions of the authors. Generally, the manuscript seems more like a generalized manuscript with no real scientific impact and just random data that does not have heads or tails in a general story about the effect of cryogenic treatment on the investigated aluminum composite. Furthermore, with a quick google search I noticed that some of the authors have already published about the topic of cryogenic treatment on the same material in Metals MDPI (https://www.mdpi.com/2075-4701/12/10/1767). To this end I would like to inquire from the authors how does the content of this manuscript differ from the recently published paper, as they seem to hold many similar results and conclusions? To be honest it seems this paper looks like a simple way of using remaining published data to get one more article out from the research rather than providing the scientific community with new understanding within the dealt topic. With all this in mind I believe the article gives no further understanding to what is already known nor is advancing beyond what the authors have presented in their previous publication.
With all the above fallacies, missing information, lack of novelty and lack of research soundness I cannot recommend this manuscript for publication.
Round 2
Reviewer 1 Report
The authors have revised a portion of the text, however, not only that they have not integrated the suggested references, they have also referred to only highly biased references, which do not represent the current state (progress) of the research of DCT on Al-alloys and composites. They have ignored the suggestion to incorporate the references and also to improve the introduction part in the way to have more scientific soundness. Furthermore, the main drawback regarding the novelty of the work and the in-depth research of the suggested topic has not been modified or advanced from the previous version. Furthermore, the manuscript and the answers on suggestions/comments, clearly indicate the lack of fundamental understanding of DCT on given material, which is again correlated to insufficient literature review of the given topic.
Additionally, since the microstructure changes are not thoroughly analyzed, but rather just supported or stated only through various references the authors do not conduct a clear investigation about what their material is like, but rather on the basis that it is similar to what others have determined and discussed. On this notion alone, this either means that the investigation is not unique and novel or that the approach is too simplistic and provides only relative information based on similarity correlation rather than providing direct confirmation through their own experiments and investigations. Furthermore, they state that the results are based on their experiments results (HRTEM/SAED) however, no results are provided from their side to support their statements. As such the whole investigation does not provide any clear view of what is the actual state of the material (microstrcture) in an objective manner nor gives a confirmation what parts of the microstructure are really composed/structured in a way as the authors state. The authors should spend more time on what the material is rather than on what is expected based on other research or previos research.
Due to this I still cannot recommend such article for publication as it at best provides only incremental advance in understanding of certain microstructural aspects in correlation to DCT, which is important for the application in industry. On these grounds, I recommend rejection of the article.